# Patterns and Determinants of Essential and Toxic Elements in Chinese Women at Mid-Pregnancy, Late Pregnancy, and Lactation

**DOI:** 10.3390/nu13020668

**Published:** 2021-02-19

**Authors:** Yubo Zhou, Lailai Yan, Hongtian Li, Xiucui Li, Yaqiong Liu, Jianmeng Liu

**Affiliations:** 1Institute of Reproductive and Child Health/Ministry of Health Key Laboratory of Reproductive Health, Peking University, Beijing 100191, China; zhouyubo@bjmu.edu.cn (Y.Z.); liht@bjmu.edu.cn (H.L.); lixiucui@bjmu.edu.cn (X.L.); 2Department of Epidemiology and Biostatistics, School of Public Health, Peking University, Beijing 100191, China; 3Department of Laboratorial Science and Technology, School of Public Health, Peking University, Beijing 100191, China; yll@bjmu.edu.cn (L.Y.); liuyaqiong@bjmu.edu.cn (Y.L.); 4Medical and Health Analysis Center, School of Public Health, Peking University, Beijing 100191, China; 5Vaccine Research Center, School of Public Health, Peking University, Beijing 100191, China

**Keywords:** essential elements, toxic elements, pregnant women, lactating women, geographic regions

## Abstract

Maternal status of essential and toxic elements affects the health of the mother, developing fetus, or breastfeeding infant. However, few studies have examined the patterns of these elements and their determinants in pregnant or lactating women. Plasma samples of 1211 healthy mid-pregnant, late pregnant, and lactating women enrolled in coastland, lakeland, and inland areas of China from May–July 2014, were analyzed for concentrations of 15 elements, using inductively coupled plasma mass spectrometry. The adjusted median concentrations of elements varied by physiologic stage and region. Lactating versus pregnant women showed higher concentrations of Zn, Cr, Mo, Ni, Sb, Cd and Pb, but lower concentrations of Cu, I, Al and Hg. In pregnant women, the concentrations of Fe, Zn, I, Mo, Ni, Al, Hg and Cd were higher in mid- versus late-pregnancy. Overall, the highest concentrations were observed in Zn, I, Mn, Al, and Pb in coastland, in Hg in lakeland, and in Fe in inland area. Element concentrations varied by maternal age, pre-pregnancy BMI, education, parity, delivery mode, feeding practice, and intakes of aquatic products and mutton. In conclusion, essential and toxic elements coexisted in pregnant and lactating women, and their concentrations varied by physiologic stages, regions, maternal socio-demographic characteristics and dietary factors.

## 1. Introduction

Essential trace elements such as zinc, copper, chromium, and manganese, acting as co-factors in enzyme reactions and playing roles in energy metabolism and cellular activities, are essential for human physiological functions [1,2]. Deficiency in these elements is relatively common among pregnant and lactating women [2], due to their increased demands. On the other hand, excessive accumulation might occur, especially when multi-micronutrient supplementations are inappropriately taken. In addition, maternal exposure to toxic elements, such as mercury, cadmium, and lead, is also concerned [3,4]. In the vulnerable periods of pregnancy and lactation, deficiency and overload of essential trace elements as well as excessive accumulation of toxic elements might co-exist, likely increasing risks of maternal complications [5,6], stillbirth [7], birth defects [8,9], fetal growth restriction [10,11,12], or long-term low neurobehavioral functions [13,14,15].

Essential and toxic elements exist in water, diet, soil, air, and commercial products, with ingestion, inhalation and dermal contact as dominant pathways of exposure. The elements’ status in pregnant and lactating women might relate to environmental, dietary, and maternal socio-demographic characteristics [16,17], suggesting that element status may differ among women from different areas. Moreover, the status may change concurrently with the rapidly changing metabolism during pregnancy and lactation [18,19]. For example, mobilization of lead from bone stores to blood likely occur alongside altered calcium metabolism during the stages [20,21].

In China, pregnant women are generally recommended to take folic acid supplements, iron-rich foods, and iodized salt at their first prenatal care visit, and to increase intake of milk, fish, poultry, eggs, and lean meat during late pregnancy. Lactating women are recommended to take iodized salt, marine products and animal source foods. A complete understanding of essential and toxic elements status in Chinese pregnant and lactating women is important to develop effective public health strategies. However, data on element concentrations in lactating women are sparse [22], and have been limited in terms of geographic coverage in pregnant women [22,23]. Given the marked diversity of geography, environment, and diet throughout China, estimates are likely to be sensitive to the area sampled. Most previous studies compared the element status between pregnant and non-pregnant women [19,23,24,25,26,27], but few focused on the comparison between pregnant and lactating women [22,28], even though the status is a continuum throughout pregnancy and lactation [2]. In addition, little is known about determinants of element status in pregnant or lactating women.

The main objectives of the present study were to describe the plasma concentrations and patterns of essential and toxic elements in pregnant and lactating women who were living in typically representative coastland, lakeland, and inland areas of China, and to identify geographic, dietary, and socio-demographic determinants of element status.

## 2. Materials and Methods

### 2.1. Study Population

A cross-sectional study was carried out in women at mid-pregnancy (15–19 weeks of gestation), late pregnancy (37–41 weeks of gestation), and lactation (35–49 days postpartum) in the Weihai, Yueyang, and Baotou cities of China from May to July 2014 [29,30,31]. Weihai, prevailing a monsoon climate with a latitude of 37°25′ N, is a coastland city located in Shandong Province of China. Yueyang, prevailing a subtropical monsoon climate with a latitude of 29°37′ N, is a lakeside city located in Hunan Province. Moreover, Baotou, prevailing a temperate continental monsoon climate with a latitude of 40°15′ N, is an inland city located in the Inner Mongolia Autonomous Region. The main manufacturing industries are agricultural and sideline food processing industry in Weihai, petrochemical industry in Yueyang, and metallurgical and rare-earth industry in Baotou.

All participants were recruited at local maternal and child health centers, when they underwent prenatal or postpartum clinic visit. Important inclusion criteria for all participants were as follows: 18 to 35 years old, the absence of comorbidities, local residents in the study areas, and willingness to comply with the study protocol. Additional inclusion criteria for pregnant women included a singleton pregnancy and no pregnancy-induced complications, and for lactating women included having a singleton live birth and currently exclusive or partial breastfeeding their infants. Among 1254 women initially recruited, 43 were excluded due to age >35 years (*n* = 23) or not being in the predetermined mid-, late-pregnancy or lactation stages (*n* = 20), leaving 1211 in the final analyses. The 1211 women were recruited approximately equally in mid-pregnancy, late pregnancy, and lactation by the three regions, with an average 135 women in each physiological period per region.

The study protocol was reviewed and approved by the Institutional Review Board /Human Subjects Committee (IRB00001052-14012; date of approval: 22 April 2014). Informed consent was obtained from all participants before their recruitment into the study.

### 2.2. Data Collection

Information about maternal socio-demographic characteristics including maternal age, ethnicity, education status, annual family income per capita, height and weight before pregnancy, parity, and gestational age at enrollment or delivery was collected using a structured questionnaire. Information about delivery mode and feeding practice was further collected for lactating women.

Dietary information in the previous month was collected using a semiquantitative, tablet-based food frequency questionnaire (FFQ) by trained obstetricians or nurses. Given that the primary aim of the original study was to assess the maternal docosahexaenoic acid (DHA) status, only DHA-specific foods were contained in the FFQ, as detailed previously [30,31]. In the present analysis, information on freshwater aquatic products (38 items, such as carp, shrimp, and river crab), marine aquatic products (75 items, such as mackerel, lobster, and crab), mutton, and formula which probably contain multiple elements was used. For each food item, a standard portion size with 9 options ranging from 25 g to 250 g, and consumption frequency with 12 options ranging from “once per month” to “three times per day” were collected. The average daily intake was calculated as multiplying the frequency of consumption in the previous month by the average amount consumed per time, and then dividing by 30 days.

### 2.3. Blood Sample Collection, Processing, and Analysis

About 5 mL of fasting venous blood was drawn from each woman at enrollment by experienced phlebotomists. Blood samples were collected into ethylene diamine tetraacetic acid-containing tubes, placed into a refrigerator at 5 °C for at least 30 min, and then processed to separate plasma aliquots within 4 h. These aliquots were stored in cryostat tubes at −20 °C at local hospitals for about 10 days, before being sent on day ice to the National Health Commission Key Laboratory of Reproductive Health at Peking University Health Science Center in China, and then stored at −80 °C until analysis.

A total of 15 elements in the plasma including copper (Cu), iron (Fe), zinc (Zn), chromium (Cr), iodine (I), manganese (Mn), cobalt (Co), molybdenum (Mo), boron (B), nickel (Ni), aluminum (Al), antimony (Sb), mercury (Hg), cadmium (Cd), and lead (Pb) were determined using inductively coupled plasma mass spectrometry (ICP-MS) (ELan DRC II; PerkinElmer, Waltham, MA, USA). A sample volume of 0.1 mL was diluted 20 times by mixing with 1.8 mL of 1% nitric acid solution and 0.1 mL of internal standards (Indium and Rhenium) in a 2-mL centrifuge tube. The operating conditions of ICP-MS for analysis were the following: 0.96 L/min atomized gas flow rate, 1.87 L/min auxiliary gas flow rate, 17.1 L/min plasma gas flow rate, 1000 W radio frequency power, 50–100 ms duration of stay, 1.1 mL/min sample lift, single point peak hopping scanning mode, and 0.7–0.9 amu resolution.

The analytical accuracy was verified by repeated analysis of Certified Reference Materials (CRMs) ClinChek^®^-Plasma Control samples (Level II, 8884). For Pb and B, standard pork liver samples (GBW10051) were used as the CRMs. The measured values of the CRMs for all elements were ranging from 95.5% to 103.0%. The limit of detection (LOD) was calculated as 3 times the standard deviation of the blank concentration. Data on CRMs and LODs are presented in Appendix A. The element analysis was completed in the Central Laboratory of Biological Elements at Peking University Health Science Center in China, with the operational procedure approved by China Metrology Accreditation.

### 2.4. Statistical Analysis

Medians and interquartile ranges (IQRs) were used to summarize the concentrations of elements because of their abnormal distribution, according to the Kolmogorov-Smirnov D tests (all *p* values < 0.01). Adjusted medians and adjusted IQRs were estimated using multivariable quantile regression models, by considering covariates including physiologic stage (mid-pregnancy, late pregnancy, and lactation), geographic region (coastland, lakeland, and inland), maternal age (≤25, 26 to 30, and 31 to 35 years), parity (nulliparous, and multiparous), ethnicity (Han, and others), education status (middle school or less, high school, and college or higher), annual family income per capita (<30,000, 30,000 to 49,999, 50,000 to 99,999, and ≥100,000 Yuan), pre-pregnancy body mass index (BMI; underweight <18.5 kg/m^2^, normal weight 18.5 to <25 kg/m^2^, and overweight/obese ≥25 kg/m^2^), and dietary intake (the lowest and middle tertiles, and the highest tertile) of aquatic products, mutton, and formula. Pre-pregnancy BMI was calculated as the pre-pregnancy weight in kilograms divided by the squared height in meters. Regarding the estimates for lactating women, delivery mode (vaginal delivery, and cesarean delivery) and feeding practice (exclusive breastfeeding, and partial breastfeeding) were additionally adjusted. The differences of adjusted median concentrations between pregnant and lactating women, or other dichotomous variables were tested using the Mann-Whitney tests; and the differences across geographic regions or other polytomous variables were tested using the Kruskal-Wallis tests, followed by Bonferroni corrected Mann-Whitney tests for multiple comparisons. In addition, multivariable quantile regression models with stepwise selection were used to examine potential determinants of elements concentrations.

All statistical analyses were performed using SPSS 24.0 software (SPSS Inc., Chicago, IL, USA). *p* values were two-sided, and statistical significance was set at *p* < 0.05.

## 3. Results

The maternal characteristics of the pregnant and lactating women analyzed in this study have been presented elsewhere. [29,31] In brief, the average (standard deviation) age of the women was 27.8 (3.0) years, and average pre-pregnancy BMI was 20.9 (2.9) kg/m^2^. Among the 1211 women, those at mid-pregnancy, late pregnancy, and lactation periods accounted for 33.6% (n = 407), 32.8% (n = 397), and 33.6% (n = 407), respectively. Women in different periods were recruited approximately equally from the coastland (32.9%), lakeland (33.2%), and inland (33.9%) regions. Most women were of Han ethnicity (95.2%), had college or higher educational status (65.7%), and had an annual per capita family income of less than 50,000 Yuan (71.4%). The average intake of freshwater aquatic products, marine aquatic products, mutton, and formula were 43.4 (88.8) g/d, 29.7 (46.0) g/d, 9.3 (42.3) g/d, and 0.7 (2.1) tsp/d, respectively (Appendix A).

Crude and adjusted median concentrations of plasma essential and toxic elements among pregnant and lactating women are presented in Table 1. For essential elements, lactating women had lower levels of Cu (1300 vs. 2025 ng/mL) and I (56.5 vs. 97.9 ng/mL), but higher levels of Zn (1064 vs. 929 ng/mL), Cr (0.43 vs. 0.38 ng/mL), and Mo (3.81 vs. 2.97 ng/mL), as compared with pregnant women. For toxic elements, lactating women had higher concentrations of Ni (16.4 vs. 12.9 ng/mL), Sb (6.42 vs. 6.03 ng/mL), Cd (1.21 vs. 0.98 ng/mL), and Pb (1.59 vs. 1.34 ng/mL), but lower concentrations of Al (65.0 vs. 75.5 ng/mL) and Hg (0.59 vs. 0.73 ng/mL).

In pregnant women, concentrations of almost all elements differed between stage of pregnancy, except Sb (Table 2). The concentrations of Fe, Zn, I, Mo, Ni, Al, Hg, and Cd were higher, while the concentrations of Cu, Cr, Mn, Co, B, and Pb were lower in mid- versus late-pregnancy. The concentrations varied significantly by geographic region, both in pregnant (Table 2) and lactating women (Table 3). The concentrations of Zn, I, Mn, Al, and Pb were highest in coastland area, followed by lakeland, and lowest in inland. The Fe concentration was highest in inland area, followed by lakeland and coastland area. The Hg concentration was highest in lakeland area, followed by coastland, and lowest in inland.

Element concentrations of pregnant and lactating women differed by maternal socio-demographic characteristics (Table 2 and Table 3). For example, in pregnant women, the concentrations of Cu and Mn increased with increasing maternal age, and the concentrations of Fe, Cr, Sb, and Cd increased with increasing pre-pregnancy BMI; in lactating women, the concentrations of Fe, Cr, Mo, and B increased with increasing maternal age, and the concentrations of Cu, Zn, Co, B, Ni, and Pb increased with increasing pre-pregnancy BMI. Pregnant women with higher versus lower education had higher concentrations of Fe and Mo, and lower concentration of Pb. In addition, the concentrations of Cu, Zn, Cr, Mo, B, Cd, and Pb varied by parity in pregnant women (Table 2), and the concentrations of Cu, Zn, Cr, I, B, Al, Ni, Sb, Hg, and Cd varied by delivery mode and feeding practice in lactating women (Table 3).

Element concentrations were also determined by dietary intakes of pregnant and lactating women (Table 4). For example, women in the highest tertile of marine aquatic product had higher concentrations of Cu, Zn, I, Mn, Co, Hg, and Pb, women in the highest tertile of mutton had a higher concentration of Fe, and women in the highest tertile of freshwater aquatic products had a higher level of Hg, as compared to those in the lowest and middle tertiles.

The multivariable quantile regression models on the association between median concentrations of elements and the potential determinants showed similar results, with physiological period and geographic region as the most influential determinants (Appendix A).

## 4. Discussion

In this large cross-sectional study conducted in coastland, lakeland, and inland regions of China, we found that the essential and toxic elements coexisted in pregnant and lactating women. Moreover, the concentrations of elements varied significantly by pregnancy and lactation stages, geographic regions, maternal socio-demographic characteristics, as well as dietary intakes.

Maternal element status is a continuum from pregnancy through lactation with varying trends [2], but few studies have explored the trends between pregnant and lactating women [22,28]. In our study, lactating women had higher concentrations of essential trace elements including Zn, Cr, and Mo, as well as toxic elements including Ni, Sb, Cd, and Pb. This finding was similar to that from North Norwegian women [28], which also found that concentrations of Zn, Mn, Cd, and Pb were higher postpartum than during pregnancy. The concentrations of Cu and I were lower in lactating women than in pregnant women in this study, similar to another Chinese study which showed that the concentration of urine I was lower in lactating than in pregnant women (194 vs. 206 μg/L) [33]. Additionally, data from New Zealand showed that the urine I concentration was consistently lower in lactating than in pregnant women before (34 vs. 47 μg/L) or after (74 vs. 85 μg/L) taking iodine supplements [34]. These findings indicate that lactating women might need more iodine to meet the demands of both the mother and the growing infant. Iodine is one of “priority” nutrients for lactating women, because maternal iodine status likely affects its amount secreted in the breast milk, subsequently affecting infants’ health [2]. The health effect of maternal iodine status on offspring warranted further study, particularly given that about 80% of infants were breastfed at 4 months with an average duration of 10 months in China [35].

Most element concentrations in pregnant women of our study were in the reference range for Chinese pregnant women derived from the China Nutrition and Health Survey (CNHS) 2010–2012 [36]. The concentrations of almost all elements varied by stages of pregnancy; the concentrations of Cu, Cr, Mn, Co, B, and Pb were higher, while the concentrations of Fe, Zn, I, Mo, Al, Ni, Hg, and Cd were lower in women at late pregnancy, compared to those at mid-pregnancy. These trends by pregnancy stages were consistent with previous studies, as maternal Cu, Mn, and Co concentrations increased while the Fe and Zn concentrations decreased as pregnancy progressed [24,36,37]. The increasing Cu and Mn concentrations were possibly due to maternal increased intestinal absorption and the increased estrogen and progesterone concentrations during pregnancy [37,38]. Cu is necessary for normal fetal hematopoiesis, and Cu deficiency during fetal development can result in structural and biochemical abnormalities [39] while high level of Cu might be linked to congenital heart defects [8]. Mn is essential for normal development and cellular function during pregnancy, but also toxic particularly neurotoxic at high exposures [40]. The decreasing concentrations of Fe and Zn were mainly due to hemodilution, fetal and placental growth, and increasing physiological requirements as pregnancy progress. Maternal zinc status is critical for the development of fetus and infant, and iron status during pregnancy is associated with pregnancy outcomes and strongly affects the iron stores of the infant at birth [2], although delayed cord clamping is recommended to improve iron stores in infancy [41]. Thus, adequate intakes of iron and zinc are important, particularly given that they are typically less well absorbed trace elements [42].

Consistent with the findings from Norwegian women [28], we observed an increasing Pb concentration from mid-pregnancy (1.27 ng/mL) to late pregnancy (1.36 ng/mL), and to lactation (1.59 ng/mL), possibly due to the mobilization of cumulative maternal Pb from bones to blood during pregnancy and lactation [20,21]. We also observed a decreasing trend in maternal Hg concentration as expected, with the highest in mid-pregnancy (0.75 ng/mL), following by late pregnancy (0.70 ng/mL), and the lowest in lactation (0.59 ng/mL), possibly because of the increasing blood volume during pregnancy and the faster blood clearance half-life of Hg during lactation [43]. Hg and Pb mainly affect central nervous system, such as impairing infant cognitive and behavioral development, and the risk effect of Pb for fetus and infant is higher and lasts longer than Hg [44]. While the relatively higher Hg status in pregnant women requires further study, given that maternal exposure to Hg is more important during fetal development than during breastfeeding [44].

The concentrations of elements differed markedly across geographic regions, possibly due to the distinct dietary patterns. Maternal Fe concentration was highest in inland compared to lakeland and coastland areas, possibly because inland women consumed more mutton (24.9 g/d vs. 0 and 2.8 g/d). As a kind of red meat, mutton likely provides considerable amounts of iron in highly bioavailable heme form [45]. Consistently, a higher Fe concentration was observed in women having more mutton in this study. Concentrations of Zn, I, and Mn were highest in coastland compared to lakeland and inland areas, possibly since women in coastland area consume more aquatic foods (98.5 g/d vs. 81.5 g/d and 39.9 g/d), particularly marine aquatic foods (62.5g/d, vs. 14.2 g/d and 12.9 g/d). Consistently, higher concentrations of Zn, I, Mn were observed in women having more marine aquatic foods. The highest concentration of Hg was observed in lakeland area, consistent to the finding of a higher Hg concentration in women having more freshwater aquatic foods.

It is challenging to make direct comparisons of element concentrations in pregnant or lactating women from different studies, because the concentrations were determined using different assay methods with various specimen types. However, some comparable studies on plasma elements for pregnant women were found in the literature. Cu status in our study was comparable to that in American women [19]. However, the level of Zn we observed was slightly higher than that of American women [19], and markedly higher than that of Korean women [24]. The concentrations of Pb and Cd were higher but the Hg concentration was lower in pregnant women in our study as compared to those in Japan [46].

This study comprehensively described the concentrations and patterns of essential and toxic elements and some of its determinants in both pregnant and lactating women, providing important information concerning element status of vulnerable populations in China. However, limitations of this study should be taken into account. Women in mid-pregnancy, late pregnancy, and lactation stages in this study were cross-sectionally selected, rather than longitudinally followed up a cohort of the population. However, the estimates, after multivariable adjustments, can largely reflect the variations in element status at different stages of pregnancy or lactation. Women in early pregnancy or non-pregnant women within the reproductive age were not included in this study, thus we could not make a complete comparison from pre-pregnancy through postpartum. Cord blood samples were not collected in the study, so fetal exposure could not be assessed. In addition, only healthy women in urban areas were enrolled in this study, therefore generalization of our findings to other pregnant or lactating women needs caution. Dietary information was not fully collected, and only data on the DHA-specific foods containing essential trace elements or toxic elements were analyzed in this study. Furthermore, although we briefly depicted the variations of element concentrations by dietary intakes, our data do not provide a bioavailable picture of the elements, which varies widely by food components, dietary interactions, and individual nutritional status [42].

## 5. Conclusions

This large cross-sectional study presented concentrations of essential and toxic elements among Chinese pregnant and lactating women, and the results were partly representing proxies for fetal and infant exposures. The concentrations differed markedly across physiologic stages, geographical regions, maternal socio-demographic characteristics, and dietary intakes. The concentration differences by physiologic periods were mainly due to the metabolism changes during the pregnancy and lactation stages, while the differences by geographic regions were mainly due to distinct dietary patterns. Our findings illustrate the coexistence of essential and toxic elements, and the complex impacts of maternal characteristics on element status, indicating that a comprehensive assessment of element status is needed before formulation of public health strategies. Additional research is needed to establish more reliable bioavailability of trace elements in pregnant and lactating women.

## Figures and Tables

**Table 1 nutrients-13-00668-t001:** Concentrations (ng/mL) of plasma essential and toxic elements among pregnant and lactating women.

Elements	Overall (*n* = 1211)		Pregnant Women (*n* = 804)		Lactating Women (*n* = 407)		*p* ^2^
	Crude Median (IQR)	Adjusted ^1^ Median (IQR)	Crude Median (IQR)	Adjusted ^1^ Median (IQR)	Crude Median (IQR)	Adjusted ^1^ Median (IQR)	
Essential elements ^3^							
Cu	1810 (1409, 2133)	1949 (1760, 2167)	2024 (1804, 2281)	2025 (1809, 2292)	1306 (1165, 1454)	1300 (1194, 1441)	<0.001
Fe	2392 (2077, 2702)	2374 (2115, 2699)	2395 (2057, 2716)	2372 (2093, 2709)	2383 (2124, 2674)	2348 (2087, 2639)	0.879
Zn	962 (843, 1101)	979 (907, 1074)	935 (816, 1077)	929 (849, 1032)	1012 (899, 1141)	1064 (974, 1166)	<0.001
I	82.1 (61.7, 104.4)	88.9 (77.3, 102.2)	96.4 (80.7, 112.5)	97.9 (84.5, 109.5)	57.5 (50.1, 66.6)	56.5 (51.2, 62.8)	<0.001
Cr	0.39 (0.31, 0.53)	0.39 (0.30, 0.51)	0.38 (0.30, 0.52)	0.38 (0.29, 0.52)	0.42 (0.33, 0.59)	0.43 (0.34, 0.55)	<0.001
Mn	2.00 (1.59, 2.64)	1.91 (1.62, 2.50)	2.03 (1.63, 2.66)	1.95 (1.71, 2.55)	1.96 (1.52, 2.63)	1.80 (1.43, 2.41)	0.059
Co	1.69 (1.32, 2.21)	1.70 (1.41, 2.12)	1.70 (1.35, 2.10)	1.67 (1.46, 1.99)	1.67 (1.26, 2.44)	1.63 (1.28, 2.39)	0.501
Mo	3.22 (2.68, 3.90)	3.22 (2.76, 3.82)	3.07 (2.58, 3.60)	2.97 (2.60, 3.35)	3.63 (3.04, 4.32)	3.81 (3.12, 4.43)	<0.001
B ^4^	65.3 (41.9, 94.4)	60.4 (42.9, 92.6)	65.7 (44.3, 88.7)	65.2 (52.5, 88.2)	64.5 (38.7, 113) b	60.1 (38.2, 101.0)	0.167
**Toxic elements**							
Ni ^5^	14.7 (9.7, 25.2)	15.8 (11.6, 24.1)	13.4 (9.1, 21.7)	12.9 (10.5, 19.6)	20.1 (11.1, 31.4)	16.4 (11.2, 29.8)	<0.001
Al	72.6 (59.9, 93.0)	71.8 (61.4, 84.6)	75.5 (61.1, 99.0)	75.5 (64.1, 88.5)	68.6 (58.5, 84.1)	65.0 (58.7, 73.5)	<0.001
Sb	6.14 (5.34, 6.98)	6.20 (5.31, 6.93)	6.04 (5.37, 6.72)	6.03 (5.52, 6.57)	6.46 (5.27, 7.58)	6.42 (5.49, 7.39)	<0.001
Hg	0.69 (0.50, 0.89)	0.68 (0.54, 0.84)	0.71 (0.52, 0.90)	0.73 (0.60, 0.89)	0.65 (0.48, 0.87)	0.59 (0.48, 0.73)	0.021
Cd	1.07 (0.85, 1.30)	1.06 (0.88, 1.27)	1.00 (0.82, 1.22)	0.98 (0.82, 1.20)	1.20 (0.99, 1.45)	1.21 (0.99, 1.45)	<0.001
Pb	1.60 (1.14, 2.23)	1.46 (1.13, 2.12)	1.53 (1.08, 2.16)	1.34 (1.06, 2.06)	1.70 (1.33, 2.45)	1.59 (1.34, 2.17)	<0.001

^1^ Estimated from multivariable quantile regression models adjusted for physiologic stage, geographic region, maternal age, parity, ethnicity, education status, annual family income per capita, pre-pregnancy BMI, and dietary intake of aquatic products, mutton, and formula; delivery mode and feeding practice were also included in the models for lactating women; ^2^ The statistical differences in the plasma element concentrations between pregnant and lactating women were tested using Mann-Whitney tests; statistical significance was set at *p* < 0.05; ^3^ Dietary reference intakes of essential elements were presented in Appendix A; ^4^ Although B was classified as an essential element, neither a recommended dietary allowance nor an adequate intake has been established for boron; ^5^ Ni is generally not considered an essential nutrient for humans at present, while evidence from experimental animals indicated the beneficial effects of Ni on metabolic processes and enzyme functions [32]; IQR, interquartile range.

**Table 2 nutrients-13-00668-t002:** Adjusted median concentrations (ng/mL) of plasma essential and toxic elements by characteristics of pregnant women (*n* = 804) ^1^.

Characteristics	*n* (%)	Essential Elements									Toxic Elements					
		Cu	Fe	Zn	I	Cr	Mn	Co	Mo	B	Ni	Al	Sb	Hg	Cd	Pb
**Pregnancy stage**																
Mid-pregnancy	407 (51)	1948 ^a^	2515 ^b^	1026 ^b^	107.3 ^b^	0.37 ^a^	1.89 ^a^	1.55 ^a^	3.02 ^b^	59.3 ^a^	13.1 ^b^	78.9 ^b^	6.04	0.75 ^b^	1.00 ^b^	1.27 ^a^
Late-pregnancy	397 (49)	2096 ^b^	2184 ^a^	895 ^a^	94.2 ^a^	0.38 ^b^	2.00 ^b^	1.91 ^b^	2.84 ^a^	73.6 ^b^	12.1 ^a^	72.9 ^a^	6.02	0.70 ^a^	0.90 ^a^	1.36 ^b^
**Geographic region**																
Coastland	263 (33)	2056 ^b^	2297 ^a^	1088 ^c^	109.9 ^c^	0.36 ^a^	2.44 ^c^	1.69 ^b^	2.87 ^a^	87.0 ^c^	13.0 ^b^	100.4 ^c^	6.03 ^b^	0.74 ^b^	0.93 ^b^	1.89 ^b^
Lakeland	267 (33)	2069 ^b^	2371 ^b^	935 ^b^	101.6 ^b^	0.39 ^b^	1.96 ^b^	1.94 ^c^	2.84 ^a^	67.3 ^b^	21.2 ^c^	75.5 ^b^	5.55 ^a^	0.85 ^c^	0.90 ^a^	1.27 ^a^
Inland	274 (34)	1993 ^a^	2518 ^c^	813 ^a^	84.1 ^a^	0.38 ^b^	1.59 ^a^	1.34 ^a^	3.58 ^b^	43.1 ^a^	9.07 ^a^	62.5 ^a^	6.55 ^c^	0.49 ^a^	1.12 ^c^	1.26 ^a^
**Age (years)**																
≤25	140 (17)	1988 ^a^	2341 ^a^	916	96.5	0.39 ^b^	1.94 ^a^	1.82	2.91	56.8 ^a^	20.1 ^b^	72.9 ^a^	5.64 ^a^	0.76	0.96	1.33
26 to 30	491 (61)	2005 ^a^	2383 ^b^	969	99.7	0.37 ^a^	1.95 ^a^	1.65	2.97	72.5 ^b^	12.5 ^a^	78.0 ^b^	6.04 ^b^	0.73	0.99	1.33
31 to 35	173 (22)	2091^b^	2356 ^a^	920	95.0	0.39 ^b^	2.06 ^b^	1.62	3.05	63.7 ^a^	13.2 ^a^	76.3 ^b^	6.03 ^b^	0.72	1.00	1.41
**Pre-pregnancy BMI**																
Underweight	147 (18)	2032 ^a^	2362 ^a^	908	97.9	0.37 ^a^	1.90	1.77 ^b^	2.91	64.5	12.7	74.8	5.90 ^a^	0.74	0.96 ^a^	1.35
Normal weight	591 (74)	2017 ^a^	2368 ^a^	957	98.5	0.38 ^a^	1.99	1.69 ^b^	2.97	67.6	13.0	76.3	6.04 ^b^	0.73	0.98 ^a^	1.33
Overweight/obese	66 (8)	2090 ^b^	2395 ^b^	914	89.3	0.43 ^b^	1.95	1.54 ^a^	3.16	59.3	11.5	75.5	6.34 ^b^	0.66	1.04 ^b^	1.37
**Parity**																
Primiparous	665 (83)	2015 ^a^	2390	968 ^b^	98.5	0.38 ^a^	1.99	1.69	2.97 ^b^	69.3 ^b^	12.8	76.3	6.04	0.73	0.99 ^b^	1.35 ^b^
Multiparous	139 (17)	2066 ^b^	2315	886 ^a^	94.2	0.39 ^b^	1.91	1.60	2.78 ^a^	56.4 ^a^	14.0	72.2	5.96	0.80	0.94 ^a^	1.27 ^a^
**Ethnicity**																
Han	764 (95)	2032 ^b^	2364 ^a^	967 ^b^	98.5 ^b^	0.38 ^b^	1.97 ^b^	1.70 ^b^	2.95 ^a^	68.1 ^b^	13.0 ^b^	76.3	6.02 ^a^	0.75	0.98 ^a^	1.35 ^b^
Others	40 (5)	1937 ^a^	2597 ^b^	845 ^a^	87.9 ^a^	0.37 ^a^	1.62 ^a^	1.28 ^a^	3.55 ^b^	40.4 ^a^	9.1 ^a^	68.7	6.89 ^b^	0.62	1.20 ^b^	1.15 ^a^
**Education**																
College or higher	523 (65)	2018 ^a^	2390 ^c^	915	94.3 ^a^	0.38 ^a^	1.95	1.62 ^a^	2.98 ^b^	64.5	12.1 ^a^	74.0 ^a^	6.09 ^b^	0.72	1.00 ^b^	1.33 ^b^
High school	175 (22)	2069 ^b^	2360 ^b^	969	105.9 ^b^	0.38 ^a^	1.95	1.91 ^b^	2.94 ^b^	60.5	13.6 ^b^	74.7 ^a^	5.85 ^a^	0.73	0.96 ^a^	1.24 ^a^
Middle school or less	106 (13)	2035 ^a^	2264 ^a^	957	98.6 ^a^	0.40 ^b^	2.04	1.62 ^a^	2.78 ^a^	69.7	13.5 ^b^	81.3 ^b^	5.93 ^a^	0.77	0.93 ^a^	1.68 ^c^
**Annual family income per capita (Yuan ^2^)**																
<30,000	296 (37)	2034 ^a^	2294 ^a^	969	98.5	0.37 ^a^	1.95	1.87 ^b^	2.91	63.0	19.7 ^b^	77.3	6.09	0.71	0.98	1.30 ^a^
30,000 to <50,000	231 (29)	2023 ^a^	2408 ^c^	985	99.8	0.38 ^a^	2.03	1.59 ^a^	2.99	75.6	13.1 ^a^	77.3	6.09	0.74	0.98	1.54 ^b^
50,000 to <100,000	191 (24)	2052 ^b^	2327 ^b^	929	97.2	0.38 ^a^	2.00	1.58 ^a^	2.97	72.5	12.4 ^a^	75.1	6.04	0.79	0.97	1.30 ^a^
≥100,000	30 (4)	2029 ^a^	2569 ^d^	928	87.7	0.42 ^b^	1.94	1.42 ^a^	2.79	73.7	12.0 ^a^	68.9	6.51	0.58	0.99	1.70 ^c^

^1^ Estimated from multivariable quantile regression models adjusted for physiologic stage, geographic region, maternal age, parity, ethnicity, education status, annual family income per capita, pre-pregnancy BMI, and dietary intake of aquatic products, mutton, and formula; delivery mode and feeding practice were also included in the models for lactating women. Values not sharing the same superscript letter (a, b, c, d) denote significant differences across characteristics of women: concentrations of a < b < c < d; statistical significance was set at *p* < 0.01 due to multiple comparisons.^2^ Eighty three women (10%) missed information on annual family income per capita.

**Table 3 nutrients-13-00668-t003:** Adjusted median concentrations (ng/mL) of plasma essential and toxic elements by characteristics of lactating women (*n* = 407) ^1^.

Characteristics	*n* (%)	Essential Elements									Toxic Elements					
		Cu	Fe	Zn	I	Cr	Mn	Co	Mo	B	Ni	Al	Sb	Hg	Cd	Pb
**Geographic region**																
Coastland	136 (33)	1335 ^b^	2333 ^a^	1106 ^c^	65.8 ^b^	0.42 ^b^	2.29 ^c^	2.43 ^c^	3.01 ^a^	114.8 ^c^	30.4 ^c^	88.5 ^c^	5.23 ^a^	0.58 ^b^	1.09 ^a^	2.16 ^c^
Lakeland	135 (33)	1246 ^a^	2316 ^a^	1074 ^b^	53.9 ^a^	0.38 ^a^	1.73 ^b^	1.31 ^a^	4.01 ^c^	40.7 ^a^	13.4 ^a^	64.1 ^b^	7.63 ^c^	0.90 ^c^	1.31 ^c^	1.57 ^b^
Inland	136 (33)	1316 ^b^	2569 ^b^	867 ^a^	54.7 ^a^	0.52 ^c^	1.64 ^a^	1.63 ^b^	3.86 ^b^	59.9 ^b^	15.8 ^b^	61.5 ^a^	6.42 ^b^	0.48 ^a^	1.25 ^b^	1.41 ^a^
**Age (years)**																
≤25	61 (15)	1189 ^a^	2242 ^a^	1054	56.6	0.40 ^a^	1.86	1.40 ^a^	3.75 ^a^	43.2 ^a^	13.3 ^a^	63.4	6.63	0.66	1.24	1.55
26 to 30	253 (62)	1324 ^b^	2347 ^b^	1067	57.0	0.42 ^a^	1.76	1.63 ^b^	3.81 ^a^	60.4 ^b^	16.8 ^b^	65.3	6.50	0.58	1.19	1.64
31 to 35	93 (23)	1298 ^b^	2364 ^c^	1072	55.3	0.45 ^b^	1.84	1.60 ^b^	4.01 ^b^	60.6 ^b^	16.6 ^b^	65.6	6.18	0.59	1.29	1.59
**Pre-pregnancy BMI**																
Underweight	80 (20)	1251 ^a^	2326	1044 ^a^	57.1	0.44	1.77	1.41 ^a^	4.00 ^b^	46.5 ^a^	14.6 ^a^	64.5	6.74 ^c^	0.62	1.31 ^b^	1.58 ^a^
Normal weight	287 (70)	1312 ^b^	2348	1072 ^b^	56.2	0.42	1.81	1.63 ^b^	3.81 ^a^	60.6 ^b^	16.4 ^b^	65.0	6.41 ^b^	0.64	1.18 ^a^	1.58 ^a^
Overweight/obese	40 (10)	1317 ^b^	2410	1079 ^b^	61.0	0.44	2.07	2.26 ^b^	3.25 ^a^	101.9 ^b^	27.3 ^b^	84.4	5.46 ^a^	0.66	1.19 ^a^	2.12 ^b^
**Ethnicity**																
Han	389 (96)	1300	2348	1067	56.8 ^b^	0.42 ^a^	1.79	1.63	3.80 ^a^	60.4	16.4	65.0	6.41	0.60	1.20 ^a^	1.58 ^a^
Others	18 (4)	1278	2357	927	50.0 ^a^	0.58 ^b^	1.95	1.58	4.57 ^b^	55.8	16.9	68.2	6.79	0.55	1.47 ^b^	2.02 ^b^
**Education**																
College or higher	273 (67)	1301	2361 ^b^	1060 ^a^	56.5	0.42	1.77	1.63	3.92	60.4	16.4	65.4	6.41	0.69	1.25 ^b^	1.59
High school	92 (23)	1283	2288 ^a^	1059 ^a^	57.3	0.43	1.85	1.56	3.67	56.6	15.6	63.4	6.27	0.60	1.19 ^b^	1.58
Middle school or less	42 (10)	1304	2232 ^a^	1100 ^b^	56.6	0.44	1.86	1.50	3.79	44.3	16.7	65.7	6.69	0.54	1.09 ^a^	1.65
**Annual family income per capita (Yuan ^2^)**																
<30,000	238 (58)	1317 ^b^	2335	1084 ^b^	56.3	0.41	1.82	1.63	3.82	60.6 ^b^	16.9 ^b^	66.3 ^b^	6.40 ^a^	0.61	1.22	1.58 ^b^
30,000 to <50,000	100 (25)	1250 ^a^	2348	1033 ^a^	55.4	0.43	1.73	1.55	3.83	60.1 ^b^	13.9 ^a^	62.3 ^b^	6.37 ^a^	0.57	1.25	1.49 ^a^
50,000 to <100,000	37 (9)	1334 ^b^	2375	1098 ^b^	56.0	0.43	1.79	1.50	3.90	48.6 ^a^	14.3 ^a^	57.3 ^a^	7.07 ^b^	0.55	1.21	1.51 ^a^
≥100,000	5 (1)	1381 ^b^	2408	1020 ^a^	61.0	0.32	1.56	1.28	4.11	41.9 ^a^	12.0 ^a^	56.7 ^a^	7.73 ^b^	0.82	1.18	1.99 ^b^
**Delivery mode**																
Vaginal delivery	243 (60)	1298	2339	1089 ^b^	60.6 ^b^	0.41 ^a^	1.98	2.33	3.27	96.4 ^b^	24.8 ^b^	67.4 ^b^	5.93 ^a^	0.61	1.16 ^a^	1.61
Cesarean delivery	164 (40)	1301	2377	934 ^a^	53.0 ^a^	0.51 ^b^	1.69	1.34	4.08	53.3 ^a^	14.1 ^a^	62.1 ^a^	6.66 ^b^	0.58	1.28 ^b^	1.58
**Feeding practice**																
Exclusive breastfeeding	240 (59)	1267 ^a^	2336	1059	55.1 ^a^	0.41 ^a^	1.82	1.75	3.89	60.4	16.7 ^b^	64.2 ^a^	6.45	0.66 ^b^	1.21	1.59
Partial breastfeeding	167 (41)	1320 ^b^	2359	1081	58.0 ^b^	0.44 ^b^	1.76	1.54	3.74	55.7	14.5 ^a^	66.2 ^b^	6.37	0.61 ^a^	1.22	1.59

^1^ Estimated from multivariable quantile regression models adjusted for physiologic stage, geographic region, maternal age, parity, ethnicity, education status, annual family income per capita, pre-pregnancy BMI, and dietary intake of aquatic products, mutton, and formula; delivery mode and feeding practice were also included in the models for lactating women. Values not sharing the same superscript letter (a, b, c) denote significant differences across groups of women: concentrations of a < b < c; statistical significance was set at *p* < 0.01 due to multiple comparisons. ^2^ Twenty seven women (7%) missed information on annual family income per capita.

**Table 4 nutrients-13-00668-t004:** Adjusted median concentrations (ng/mL) of plasma essential and toxic elements by dietary intakes of pregnant and lactating women (*n* = 1208) ^1^.

Dietary Intakes	*n* (%)	Essential Elements									Toxic Elements					
		Cu	Fe	Zn	I	Cr	Mn	Co	Mo	B	Ni	Al	Sb	Hg	Cd	Pb
**Pregnant women**																
**Freshwater aquatic product**																
Lowest and middle tertiles	580 (72)	2020	2345	931 ^a^	97.3 ^a^	0.37 ^a^	1.97 ^b^	1.62 ^a^	3.05 ^b^	69.1	12.6 ^a^	75.6 ^b^	6.10 ^b^	0.68 ^a^	1.00 ^b^	1.43 ^b^
Highest tertile	223 (28)	2036	2395	943 ^b^	99.2 ^b^	0.38 ^b^	1.94 ^a^	1.87 ^b^	2.87 ^a^	63.0	21.0 ^b^	75.1 ^a^	5.55 ^a^	0.83 ^b^	0.93 ^a^	1.26 ^a^
**Marine aquatic product**																
Lowest and middle tertiles	494 (62)	2015 ^a^	2398 ^b^	892 ^a^	92.6 ^a^	0.38 ^b^	1.86 ^a^	1.67 ^a^	3.05 ^b^	57.5 ^a^	13.8 ^b^	72.4 ^a^	6.07	0.70 ^a^	1.00 ^b^	1.30 ^a^
Highest tertile	309 (38)	2054 ^b^	2296 ^a^	1000 ^b^	102.5 ^b^	0.37 ^a^	2.32 ^b^	1.70 ^b^	2.87 ^a^	80.6 ^b^	13.0 ^a^	94.7 ^b^	6.03	0.74 ^b^	0.94 ^a^	1.84 ^b^
**Mutton**																
Lowest and middle tertiles	531 (66)	2037 ^b^	2339 ^a^	994 ^b^	102.5 ^b^	0.37 ^a^	2.01 ^b^	1.86	2.89 ^a^	72.9 ^b^	19.7 ^b^	78.4 ^b^	5.78 ^a^	0.79 ^b^	0.94 ^a^	1.38 ^b^
Highest tertile	272 (34)	2013 ^a^	2386 ^b^	853 ^a^	86.9 ^a^	0.38 ^b^	1.64 ^a^	1.49	3.43 ^b^	48.0 ^a^	9.2 ^a^	65.2 ^a^	6.44 ^b^	0.52 ^a^	1.06 ^b^	1.30 ^a^
**Formula**																
No	610 (76)	2005 ^a^	2366	937	98.3	0.38	1.90 ^a^	1.62 ^a^	3.05 ^b^	62.3 ^a^	12.5 ^a^	75.3	6.09 ^b^	0.68 ^a^	1.00 ^b^	1.39 ^b^
Yes	194 (24)	2109 ^b^	2330	933	97.6	0.38	2.01 ^b^	1.87 ^b^	2.87 ^a^	71.8 ^b^	20.8 ^b^	75.6	5.61 ^a^	0.81 ^b^	0.91 ^a^	1.30 ^a^
**Lactating women**																
**Freshwater aquatic product**																
Lowest and middle tertiles	231 (57)	1314	2417 ^b^	1055	56.4	0.45 ^b^	1.82	1.63 ^b^	3.75 ^a^	60.8	16.9 ^b^	63.3 ^a^	6.24 ^a^	0.55 ^a^	1.19 ^a^	1.62
Highest tertile	174 (43)	1303	2313 ^a^	1061	56.5	0.40 ^a^	1.85	1.48 ^a^	3.87 ^b^	53.7	15.7 ^a^	66.7 ^b^	6.71 ^b^	0.69 ^b^	1.24 ^b^	1.65
**Marine aquatic product**																
Lowest and middle tertiles	310 (77)	1301 ^a^	2375 ^b^	1048 ^a^	55.0 ^a^	0.43	1.78 ^a^	1.51 ^a^	3.87 ^b^	54.7 ^a^	15.4 ^a^	64.0 ^a^	6.50 ^b^	0.59	1.23 ^b^	1.57 ^a^
Highest tertile	95 (23)	1329 ^b^	2297 ^a^	1075 ^b^	64.9 ^b^	0.42	2.16 ^b^	2.36 ^b^	3.13 ^a^	107.7 ^b^	28.1 ^b^	87.9 ^b^	5.27 ^a^	0.61	1.13 ^a^	2.10 ^b^
**Mutton**																
Lowest and middle tertiles	282 (70)	1292 ^a^	2321 ^a^	1077 ^b^	58.1 ^b^	0.40 ^a^	1.90 ^b^	1.47 ^a^	3.77 ^a^	54.9	16.4	67.0 ^b^	6.61 ^b^	0.67 ^b^	1.19	1.71 ^b^
Highest tertile	123 (30)	1339 ^b^	2512 ^b^	894 ^a^	54.2 ^a^	0.52 ^b^	1.64 ^a^	1.62 ^b^	3.85 ^b^	60.1	16.8	62.2 ^a^	6.24 ^a^	0.50 ^a^	1.23	1.47 ^a^
**Formula**																
No	356 (87)	1307	2352	1064	56.9	0.43 ^b^	1.89 ^b^	1.64 ^b^	3.75 ^a^	61.0 ^b^	17.1 ^b^	65.6 ^b^	6.22 ^a^	0.59 ^a^	1.19 ^a^	1.67 ^b^
Yes	51 (13)	1299	2397	1035	54.9	0.41 ^a^	1.61 ^a^	1.35 ^a^	4.07 ^b^	44.0 ^a^	13.5 ^a^	62.6 ^a^	7.66 ^b^	0.84 ^b^	1.28 ^b^	1.53 ^a^

^1^ Estimated from multivariable quantile regression models adjusted for physiologic stage, geographic region, maternal age, parity, ethnicity, education status, annual family income per capita, pre-pregnancy BMI, and dietary intake of aquatic products, mutton, and formula; delivery mode and feeding practice were also included in the models for lactating women. Values not sharing the same superscript letter (a, b) denote significant differences across characteristics of women: concentrations of a < b. A total of 1208 participants were included in the analyses for dietary intakes, because 3 missed dietary information.

## Data Availability

The data presented in this study are available on request from the corresponding author. The data are not publicly available due to containing identifiable personal information.

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
