# Peer review of "Patterns and Determinants of Essential and Toxic Elements in Chinese Women at Mid-Pregnancy, Late Pregnancy, and Lactation"

_nutrients, 2021, doi:10.3390/nu13020668_

Round 1
Reviewer 1 Report
The authors have made substantial efforts changing the manuscript and have answered to all queries from the reviewers. However, as the authors state, the aim of the study is to describe concentrations and patterns of essential and toxic elements, but it is also to identify determinants of element status, as it appears also in the title.
The article as it is, it is only descriptive and therefore it does not answer the main question of the study. My comments in the first review about this (“The study it would be of higher interest if beside being descriptive, the authors analyse the weight of each variable in pregnant and lactating women. It is important to identify which factors influence and how much they influence”) has been answered, but not performed the analysis.
I believe it is necessary, for the manuscript to be interesting to publish, that the authors perform a multiple regression, possibly with a stepwise selection to be able to identify which factors determine the element status and patterns.
Reviewer 2 Report
I have no further comments
Reviewer 3 Report
This cross-sectional study was carried out on 1211 Chinese women in mid- and late pregnancy and in the lactation period. The measurements of 15 elements including essential and toxic elements was carried out and the variations of these concentrations in three different regions are presented. It is an interesting, well-written study. However, I have a few comments to increase the representability of these results:
- In the introduction section the authors should explain what are the sources of these essential trace and toxic elements in the nature? Are any of these contained in dietary products?
- Page 2, line 53: I agree with the authors that the understanding of these elements is essential to develop public health strategies. Could the authors discuss further about the dietary recommendations pregnant women receive in China at the preconception visit?
- Page 2, lines 68-80: the authors should provide more detailed description of three different regions included in the study. What weather conditions prevail in these regions? What kind of manufacturing is present in these regions?
- The study was conducted ‘from May to July 2014 as discussed previously’? I think it is not appropriate to state ‘as described previously’ and that the methodology of the original study should be presented in greater detail.
- The lactation period in the study is defined as 35-49 days post partum. Please add the information (preferably in the discussion) on what percentage of women breastfeed in China and what is the average duration of breastfeeding in China.
- Why did the authors conduct an analysis almost 7 years after the end of the study?
- What does ‘general good health determined by physicians’ mean? Would it be better if the authors wrote: the absence of comorbidities?
- The discussion should be more balanced. The authors should discuss why they measured these specific elements in pregnancy and postpartum? What is the possible physiologic explanation for the change of these concentrations?
- Discussion, line 30-33: the authors state that maternal iron stores affect the iron stores of the infant at birth. Please provide further discussion about the role of delayed cord clamping on this matter.
Round 2
Reviewer 1 Report
The authors have made the changes according to the reviewer suggestions and I believe the article has improved substantially.
I think the authors should add or change in the abstract the sentence of conclusion to state the variables that are more influential.
And the same for the conclusions in the text.
Reviewer 3 Report
All questions have been properly answered and I believe the manuscript is now suitable for publication
This manuscript is a resubmission of an earlier submission. The following is a list of the peer review reports and author responses from that submission.
Round 1
Reviewer 1 Report
The paper is clear and well written.
I have the following concerns:
- More details about the FFQ should be given: how many items? Has it been validated?
- No data about total caloric intake and concentrations of essential and toxic elements is available. Are there any undernourished patients among the studied women? Is there a correlation between energy intake and blood levels of these elements?
- Similarly, is there any correlation between overall dietary habits and levels of the elements (i.e., is a poor diet associated with a lower level of essential elements?)
Reviewer 2 Report
The authors analyse the essential and toxic elements in women in different stages such as pregnancy and lactating. The study it would be of higher interest if beside being descriptive, the authors analyse the weight of each variable in pregnant and lactating women. It is important to identify which factors influence and how much they influence. For example, to know if the differences found between concentrations of elements, are due to differences in the dietary patterns or other influencing factors. The authors point it out in the discussion, however they can analyse it since they have the FFQ data.
I would appreciate if the authors can explain more about the average BMI in pre-pregnant women and its differences with international standard values. It seems pretty low, but it can be due to the characteristics of the population.
In the description of the methodology, the authors describe the FFQ as a semiquantitative questionnaire. This needs to be better explained since the FFQ is a quantitative measure.
Further, in the discussion and conclusions, results are just described. It is necessary to explain the impact and importance of those findings and its implications.
Reviewer 3 Report
The authors provide an interesting manuscript regarding dietary minerals during various stages of pregnancy and lactation.
There are many grammatical errors, which should be addressed by the authors and editors.
The authors are cautioned not to imply or use words such as prevent, cure, diagnose, mitigate or treat within the text. These words are reserved for drugs under US law. The only exceptions are preventable diseases associated with frank deficiencies, such as vitamin C and scurvy, vitamin A and night blindness.
Abstract:
The authors really do not attempt to address the clinical relevancies of these minerals, and only make glib remarks on higher and lower concentrations. For example, the authors note Ni without commenting on the importance of Ni in an array of enzyme systems both in humans and microbes (e.g., gut microbiome).
Introduction (lines 34-64)
Line 37 needs a reference pertinent to mineral deficiencies or inadequacies during pregnancy and lactation. WHO has generated a number of excellent position papers on this topic.
Line 41 toxic elements are not defined; it's important to recognize that all elements, even those which are required, may be toxic when consumed above the upper limits. In addition, we must remember that exposure to some "toxic" elements does not necessarily trigger adverse events.
Line 60 why the abrupt change to vitamin A deficiency (not a mineral)? BTW, there is considerable information on vitamin A and pregnancy, such as vitamin A supplementation during pregnancy located on the WHO website.
The top focus on insufficiency should include iodine, one of the top three nutrient deficiencies noted the WHO action plan.; the iodine noted in Table 1 is good. This table suggests B (boron) is essential; has the Chinese government determined a RDA or DV for this mineral? While the research data on boron inadequacies among zebra fish are interesting, the translation to humans has not been accepted by the National Academies of Science in the US. The only reference to boron advanced by NAS is that 20 mg/d is an upper limit (2001 report). Due to insufficient or inconsistent data, neither an RDA nor an AI was established for boron.
Materials & Methods (Lines 65-125)
Generally, this section is fine.
Line 91 - why the sudden change that attempts to include DHA? The identified foods have minimal concentrations of DHA. Are these foods consumed to significant quantities as determined by the FFQ?
Statistics (lines 126-148)
The statistical approaches are reasonable.
Results (lines 149- 188)
Table 1 should include the DRI, RDA or DV for the respective essential elements. These recommendations could come from the Chinese government or other country, such as the US. Importantly, the authors should comment about bioavailability of the identified minerals. The uptake of these minerals varies widely based on the food chemistry and their innate composition. Thus, analytical data or data based on FFQ do not provide a biological picture of the potential health or toxicity issues.
While the authors included Ni as a toxic element, they should acknowledge its importance in an array of metabolic processes and enzyme functions. The authors are referred to a recent paper by Nielsen (Adv Nutr 2020;00:1–2; doi: https://doi.org/10.1093/advances/nmaa154)
Table 2 is interesting. Recommend at least a footnote on Ni based on the authors' data and comments from Nielsen. Similar comments may be warranted in tables 3 & 4.
Discussion (Lines 1-70 --- why were the lines renumbered)
Basically, this section discussed high/low dietary concentrations (what is high vs low) of the identified minerals. The clinical merits of the results are not well presented. What biomarkers are important during pregnancy and lactation? The comments (line 20-37) are too glib. The physiological significance of the authors' findings is missing in this section.
While the results are interesting, the biological assessments, especially in the absence of bioavailability and relevant toxicology, are without clinical value.
Conclusion (lines 71-81)
Where are the study weaknesses and strengths?
Again, the authors fail to comment on bioavailability factors that could translate to or impact pregnancy outcomes or human milk properties.
The closing remarks (lines 79-80) are classic fillers. The authors do not purpose any clinical significance or clearly advance next steps germane to this at risk population (pregnant women, lactating women, or their children especially during the first 1000 days of life).